Phylogenomic analysis and revised classification of atypoid mygalomorph spiders (Araneae, Mygalomorphae), with notes on arachnid ultraconserved element loci

Hedin Marshal mhedin@sdsu.edu 1
Derkarabetian Shahan 1 2 3
Alfaro Adan 1
Ramírez Martín J. 4
Bond Jason E. 5
1 Department of Biology, San Diego State University , San Diego , CA , United States of America
2 Department of Biology, University of California, Riverside , Riverside , CA , United States of America
3 Department of Organismic and Evolutionary Biology, Museum of Comparative Zoology, Harvard University , Cambridge , MA , United States of America
4 Division of Arachnology, Museo Argentino de Ciencias Naturales “Bernardino Rivadavia”, Consejo Nacional de Investigaciones Científicas y Técnicas (CONICET) , Buenos Aires , Argentina
5 Department of Entomology and Nematology, University of California , Davis , CA , United States of America
Gillespie Joseph
Electronic publication date: 2019 May 3
Publication date: 2019
Volume: 7
Electronic Location ID: e6864
Received 2019 Feb 1; Accepted 2019 Mar 28
Copyright: ©2019 Hedin et al.
Copyright year: 2019
Copyright holder: Hedin et al.
License: This is an open access article distributed under the terms of the Creative Commons Attribution License, which permits unrestricted use, distribution, reproduction and adaptation in any medium and for any purpose provided that it is properly attributed. For attribution, the original author(s), title, publication source (PeerJ) and either DOI or URL of the article must be cited.
License URL: https://creativecommons.org/licenses/by/4.0/

Keywords: Spider, Phylogenomics, Ultraconserved element, Orthology, Web evolution, Mygalomorphae, Taxonomy, Cryptic species, Exons

Funding: US National Science Foundation DEB 1354558 Research was funded by US National Science Foundation grants to Marshal Hedin (DEB 1354558). The funders had no role in study design, data collection and analysis, decision to publish, or preparation of the manuscript.

==============================
The atypoid mygalomorphs include spiders from three described families that build a diverse array of entrance web constructs, including funnel-and-sheet webs, purse webs, trapdoors, turrets and silken collars. Molecular phylogenetic analyses have generally supported the monophyly of Atypoidea, but prior studies have not sampled all relevant taxa. Here we generated a dataset of ultraconserved element loci for all described atypoid genera, including taxa (Mecicobothrium and Hexurella) key to understanding familial monophyly, divergence times, and patterns of entrance web evolution. We show that the conserved regions of the arachnid UCE probe set target exons, such that it should be possible to combine UCE and transcriptome datasets in arachnids. We also show that different UCE probes sometimes target the same protein, and under the matching parameters used here show that UCE alignments sometimes include non-orthologs. Using multiple curated phylogenomic matrices we recover a monophyletic Atypoidea, and reveal that the family Mecicobothriidae comprises four separate and divergent lineages. Fossil-calibrated divergence time analyses suggest ancient Triassic (or older) origins for several relictual atypoid lineages, with late Cretaceous/early Tertiary divergences within some genera indicating a high potential for cryptic species diversity. The ancestral entrance web construct for atypoids, and all mygalomorphs, is reconstructed as a funnel-and-sheet web.

Introduction

Phylogenetic evidence now overwhelmingly indicates that the mygalomorph spiders, including trapdoor spiders and their kin, are divided into the primary clades Avicularioidea and Atypoidea (Hedin & Bond, 2006; Bond et al., 2012; Hamilton et al., 2016; Garrison et al., 2016; Wheeler et al., 2017; Hedin et al., 2018a; Fernández et al., 2018; Opatova et al., 2019). Avicularioidea includes the most familiar mygalomorphs (e.g., tarantulas), and the bulk of known taxonomic diversity (World Spider Catalog, 2019). Phylogenomic analyses based on sequence-capture data have now dramatically changed our understanding of family-level diversity and interrelationships within the avicularioids (Hamilton et al., 2016; Hedin et al., 2018a; Opatova et al., 2019), with many families previously suspected of non-monophyly now known to constitute multiple independent lineages (Hedin et al., 2018a; Opatova et al., 2019).

Figure 1 Images of live animals and entrance web constructs.

(A) Hexurella apachea, Cochise County, AZ. MCH 18_029. (B) Mecicobothrium thorelli, image by G. Pompozzi. (C) Atypus karschi. Honshu, Tottori, Japan. MCH 15_016. (D) Megahexura fulva, Fresno County, CA. MCH 09_018. (E) Aliatypus californicus. Contra Costa County, CA. MCH 10_031. (F) Hexura picea. Lincoln County, OR. MCH 14_040. (G) Antrodiaetus unicolor, Jackson County, NC. (H) Atypoides (= Antrodiaetus) riversi, San Mateo County, CA. MCH 10_015. Arrows point to dorsal abdominal tergites in images B, E and G. All photos (other than Mecicobothrium) by M. Hedin.

Avicularioids are sister to Atypoidea, the latter group representing an old taxonomic hypothesis (Simon, 1892). Atypoidea was first suggested then refuted by morphology, then supported by few-gene molecular studies, and is now seemingly confirmed by phylogenomic approaches. This clade, sometimes referred to as the “atypical tarantulas” (Gertsch, 1949), includes three described families (Antrodiaetidae, Atypidae, Mecicobothriidae) whose members possess dorsal abdominal tergites (Figs. 1B, 1E and 1G). These tergites are believed to represent the vestiges of abdominal segmentation, as found in spider relatives and early-diverging spiders. Adult male atypoids possess a palpus with a conductor, females have bipartite spermathecal organs, and members of both sexes typically possess six spinnerets (Eskov & Zonstein, 1990). This clade is relatively ancient, as multiple fossil genera placed within the three described families are known from the Lower Cretaceous (100–112 MYA) of Mongolia (Eskov & Zonstein, 1990). Dalla Vecchia & Selden (2013) placed the Triassic (210–215 MYA) Friularachne into Atypoidea, but left the family-level placement unspecified.

Atypoids utilize silk to build many different types of burrow entrance constructs (Coyle, 1986). The mecicobothriid genera Mecicobothrium, Megahexura, Hexura, and Hexurella are all ground-dwelling spiders found living under objects or in earthen crevices, using elongate spinnerets to build silken funnel-and-sheet webs (Gertsch & Platnick, 1979; Costa & Pérez-Miles, 1998; pers. obs.; Figs. 1A, 1B, 1D and 1F). The atypid genera either live in subterranean burrows with open silk-lined entrances (Calommata), or build cryptic silken capture tubes extending horizontally or vertically from burrow entrances (all atypid genera, Schwendinger, 1990; Fourie, Haddad & Jocqué, 2011; Fig. 1C). Finally, the antrodiaetids live in subterranean burrows with silken turret or collapsible collar entrances (Antrodiaetus), or build trapdoors to cover their burrows (Aliatypus, Coyle, 1971; Figs. 1E, 1G and 1H). Most atypoid taxa are distributed on northern continents, although Mecicobothrium occurs in southern South America, and Calommata species are found in east Asia and throughout sub-Saharan Africa (World Spider Catalog, 2019).

Faircloth et al. (2012) first used the sequence capture of ultraconserved elements (UCEs) in phylogenomic analyses of various amniote lineages. In vertebrates more generally, core UCE regions show extreme sequence conservation, making design of broad-utility nucleotide baits possible (e.g., for all fishes, all amniotes, etc.). The function and genomic position of vertebrate UCEs has remained somewhat elusive, although most are believed to have regulatory functions and lie outside of exons (e.g., Bejerano et al., 2004; Polychronopoulos et al., 2017; McCole et al., 2018). More recently, UCE baits have been designed for megadiverse arthropod lineages, including arachnids and multiple insect orders (Faircloth et al., 2015; Faircloth, 2017). Bossert & Danforth (2018) showed that a universal set of 100 UCEs are shared across all arthropods, and that these “core” UCEs are entirely or partially exonic in origin, thus differing from vertebrate UCEs. In this paper we further explore the function of genomic regions captured by the arachnid UCE bait set. This set was tested in situ by Starrett et al. (2017), and has been used in multiple phylogenomic studies (Derkarabetian et al., 2018a; Hedin et al., 2018a; Hedin et al., 2018b; Wood et al., 2018). Knowing the functional role of arachnid UCEs has clear importance in phylogenomic analyses, potentially impacting sequence alignment, model selection, data partitioning, detection of paralogy, and so on. This is particularly true in a lineage such as spiders, where an ancient whole-genome duplication event has occurred (Clarke et al., 2015; Schwager et al., 2017), perhaps complicating orthology assignment.

Interrelationships within Atypoidea have varied considerably in past molecular phylogenetic studies (Fig. 2), and no prior studies have simultaneously sampled all known (described) atypoid genera. Here we present such an analysis with all genera, including key taxa such as Mecicobothrium and the diminutive Hexurella, neither included in prior molecular phylogenetic analyses. Using an annotated UCE locus set with BLAST evidence for gene function and orthology, we demonstrate that Atypoidea is monophyletic, while revealing multiple cases of non-monophyly within described families. Early-diverging atypoid lineages are often species-poor (approximating monotypic) and use silk to build funnel-and-sheet webs, while more diverse silken constructs have evolved in derived atypoid lineages. Similar patterns of species and web diversification occur in parallel in the avicularioids (Opatova et al., 2019).

Materials & Methods

Taxon sampling

Representatives of all nine described atypoid genera (World Spider Catalog, 2019) were sampled. Within genera, the sample included all three known species in the synonymized genus Atypoides (species now included in Antrodiaetus, (Hendrixson & Bond, 2007), two species of originally-described Antrodiaetus which span the hypothesized root node of this taxon (Hendrixson & Bond, 2007; Hendrixson & Bond, 2009), two species of Aliatypus which span the hypothesized root node of this genus (Satler et al., 2011), both described Hexura species, two geographically separated species of Hexurella, two geographically distant populations of the monotypic Megahexura fulva, and two species of the genus Sphodros. Only Mecicobothrium, Calommata and Atypus were represented by single specimens (Table S1). To confirm atypoid monophyly we sampled a handful of representative avicularioid taxa, including genera representing multiple early-diverging avicularioid lineages (Bond et al., 2012; Hedin et al., 2018a; Opatova et al., 2019). Mygalomorphs are sisters to araneomorph spiders—we used an early-diverging araneomorph lineage (Hypochilus) to root trees. In total, we gathered original UCE data for 15 specimens; data for 12 specimens were taken from previous studies (Starrett et al., 2017; Hedin et al., 2018a; Table S1). Permits for the collection of Australian specimens were granted by the Queensland Environmental Protection Agency (permit #WISP01242003).

Figure 2 Summary of previous molecular phylogenetic analyses including members of Atypoidea.

References as in text.

DNA extraction

Most specimens were preserved for DNA studies (preserved in high percentage ethyl alcohol at -80C), and genomic DNA was extracted from leg tissue using the Qiagen DNeasy Blood and Tissue Kit (Qiagen, Valencia, CA). For a handful of tissues preserved in 70–80% we used either standard phenol/chloroform extractions with 24-hour incubation for lysis, or used a modification of the Tin, Economo & Mikheyev (2014) protocol (Table S1). All extractions were quantified using a Qubit Fluorometer (Life Technologies, Inc.) and quality was assessed on agarose gels. Between 22–500 ng total DNA was used for UCE library preparation (Table S1).

UCE data collection & matrix assembly

UCE data were collected in multiple library preparation and sequencing experiments. Up to 500 ng of genomic DNA was used in sonication, using a Covaris M220 Focused-ultrasonicator. Library preparation followed methods previously used for arachnids as in Starrett et al. (2017), Derkarabetian et al. (2018a), Derkarabetian et al. (2018b), Hedin et al. (2018a) and Hedin et al. (2018b). Target enrichment was performed using the MYbaits Arachnida 1.1K version 1 kit (Arbor Biosciences; Faircloth, 2017) following the Target Enrichment of Illumina Libraries v. 1.5 protocol (http://ultraconserved.org/#protocols). Libraries were sequenced on an Illumina HiSeq 2500 (Brigham Young University DNA Sequencing Center).

Raw demultiplexed reads were processed with the Phyluce pipeline (Faircloth, 2016). Quality control and adapter removal were conducted with the Illumiprocessor wrapper (Faircloth, 2013). Assemblies were created with Velvet (Zerbino & Birney, 2008) and/or Trinity (Grabherr et al., 2011), both at default settings. When contigs from both assemblies were available, these were combined for probe matching, retrieving assembly-specific UCEs and overall increasing the number of UCEs per sample relative to using only a single assembly method. Contigs were matched to probes using minimum coverage and minimum identity values at liberal values of 65. UCE loci were aligned with MAFFT (Katoh & Standley, 2013) and trimmed with Gblocks (Castresana, 2000; Talavera & Castresana, 2007), using –b1 0.5 –b2 0.5 –b3 6 –b4 6 settings in the Phyluce pipeline.

UCE locus annotation, Matrix filtering, and Phylogenomic analyses

698 loci were found in a Phyluce 70% occupancy matrix. For the consensus sequence from each locus alignment, BLAST X searches in Geneious 10.1 (Biomatters Ltd.) were conducted against a local database (max e value of 1 × 10−10) comprising protein sequences for four arachnid taxa: Limulus polyphemus (https://www.ncbi.nlm.nih.gov/genome/787), Ixodes scapularis (https://www.ncbi.nlm.nih.gov/genome/?term=523), Stegodyphus mimosarum (https://www.ncbi.nlm.nih.gov/genome/?term=12925) and Parasteatoda tepidariorum (https://www.ncbi.nlm.nih.gov/genome/?term=13270).

BLAST annotation indicated that essentially all spider UCE loci are either entirely exonic, or exons with flanking introns (see Results). This annotation information allowed us to further curate Phyluce alignments in several ways. First, we discovered that some individual loci were part of the same protein, likely exons (or parts thereof) separated by introns (see Results). Second, annotation indicated that some UCE loci could potentially include paralogs of the same protein, or orthologs of two or more different proteins. We thus visually inspected all UCE locus alignments and excluded loci with non-orthology as evidenced by congeneric taxa with divergent sequences, using RAxML gene trees (see below) to confirm this non-orthology. Finally, annotation allowed us to define exon/intron boundaries, and exclude a majority of intron sequence for some analyses.

Three matrices were assembled for phylogenomic analyses, including (1) 70% occupancy Phyluce unfiltered (including some protein duplicates, some loci with non-orthologs), (2) 70% exon + intron, no “paralogs”, retaining one UCE locus from a set including duplicates (alignment with most sequences, or longest alignment if approximately same number of taxa), (3) 70% filtered as #2 above, plus using stricter Gblocks settings (–b1 0.5 –b2 0.85 –b3 4 –b4 8) to further trim alignments. We visually checked to confirm that these trimmed alignments comprised mostly exon data. Unpartitioned and partitioned concatenated maximum likelihood analyses were run for each dataset above. Unpartitioned analyses were conducted with RAxML version 8.2 (Stamatakis, 2014) using a GTRGAMMA model and 200 rapid bootstrap replicates. Partitioned maximum likelihood analyses were conducted using IQ-TREE (Nguyen et al., 2015; Chernomor, Von Haeseler & Minh, 2016) with partitions and models determined using ModelFinder (Kalyaanamoorthy et al., 2017), and support estimated via 1000 ultrafast bootstrap replicates (Hoang et al., 2018). Finally, we used SVDquartets (Chifman & Kubatko, 2014; Chifman & Kubatko, 2015) with n = 500 bootstraps, as implemented in PAUP* 4.0a163 (Swofford, 2003).

Web evolution and divergence time analysis

Mesquite version 3.51 (Maddison & Maddison, 2018) was used to reconstruct ancestral states for entrance web constructs, with tip values scored as seven different discrete states. Tip scorings were derived from published literature (references in Introduction), supplemented with original observation. Maximum likelihood reconstructions were produced using the one-parameter Markov k-state model (Lewis, 2001), using the RAxML exon only topology as input.

We estimated absolute divergence times using the lognormal relaxed clock model (Thorne, Kishino & Painter, 1998) implemented in Phylobayes 4.1c (Lartillot & Philippe, 2004). We used the exon only matrix, with the RAxML topology as a constraint tree. Four MCMC chains were run in parallel, stopping after 30,000 points. Analyses were checked for convergence, and considered converged when the largest discrepancy observed across bipartitions (maxdiff) was equal to 0. Posterior estimates of ages and highest posterior density (HPD) values were summarized on a single target tree from all input trees using TreeAnnotator (Bouckaert et al., 2014). Three fossil calibrations were used, with a soft bounds model (Yang & Rannala, 2006) and a birth death prior on divergence times, as follows: (1) minimum age for the root node of mygalomorphs = 240 MYA, based on Rosamygale, the oldest known mygalomorph fossil (Selden & Gall, 1992). This taxon was placed by original authors as an avicularioid, but is treated more conservatively here. (2) minimum age for the root node of Atypoidea = 210 MYA, based on Friularachne (Dalla Vecchia & Selden, 2013). We also used an alternative second calibration, using the Eskov & Zonstein (1990) fossils Ambiortiphagus and Cretacattyma to set the minimum age for the most recent common ancestor of Atypidae and Antrodiaetidae at 100 MYA. (3) minimum age for the root node of Avicularioidea = 216 MYA, based on Edwa (Raven, Jell & Knezour, 2015), a likely early-diverging avicularioid. For all three calibrations we used a maximum age of 390 MYA, corresponding to the age of fossil Uraraneida, the putative sister group of spiders (Selden, Shear & Sutton, 2008). This approximate age is in accord with maximum dates derived from other molecular clock analyses of spiders (Ayoub et al., 2007; Wood et al., 2012; Starrett et al., 2013; Fernández et al., 2018; Opatova et al., 2019).

Results

Voucher data, input DNA values, assembled contig numbers, and UCE locus numbers are found in Table S1. Except for museum samples of Mecicobothrium, all samples returned multiple 100s of loci for all matrices. We highlight Mecicobothrium—although we are confident in the results presented here (based on identical placement across all analyses), future studies with fresh specimens should verify the phylogenetic placement discussed below. Raw reads from fifteen original samples have been submitted to the SRA (SAMN10839235—10839249).

Annotation of the ∼700 loci derived from the Phyluce pipeline indicates that spider UCEs are primary exonic in origin, as essentially all (>98%) alignments BLAST to proteins found in Stegodyphus and Parasteatoda spiders, with relatively high BIT scores (Tables S2 and S3). We note that Stegodyphus and Parasteatoda are true spiders in the clade sister to mygalomorphs; we did not conduct custom BLAST searches against mygalomorphs, as the only sequenced genome (Acanthoscurria) is low coverage and incomplete (Sanggaard et al., 2014). Even the handful of UCE loci without BLAST hits contained open reading frames of variable length, and these could represent proteins that are particularly divergent from araneomorphs, or restricted to mygalomorphs.

We found that 112 total alignments mapped to the same 74 proteins (i.e., different alignments hit same protein; Tables S2 and S3). We confirmed that the conserved regions of these separate alignments represented different exons of typically large proteins, and that these exons are likely separated by very long introns (using the known short exon- long intron structure of spiders as models, see Sanggaard et al., 2014). BLAST and visual assessment of the 70% Phyluce matrices indicated that 106 alignments included non-orthologous sequences, and this was confirmed via RAxML analysis of these individual alignments (.tre files in Data S1). Non-orthology was also indicated by annotation, as most alignments including “paralogs” hit two or more different proteins at similar BIT score values (Table S2). The issue of non-orthology is further discussed below. The final matrices were populated as follows: (1) Phyluce unfiltered 70% occupancy (698 loci, 191,855 basepairs), (2) 70% filtered exon + intron (480 loci, 137,170 basepairs), (3) 70% filtered exon only (480 loci, 71,483 basepairs). All aligned matrices and .tre files are available in Data S1.

Except for one node, all nine phylogenomic analyses recover an identical branching topology within Atypoidea, albeit with variation in branch lengths and node support (Fig. 3). The single node in question involves the interrelationships of Antrodiaetus riversi, A. gertschi, and A. hadros, all previously in the synonymized genus Atypoides. Overall, the following pertinent clades were recovered with high support (bootstrap >95 and posterior probability >0.95) in all analyses: Avicularioidea, Atypoidea, Atypidae, and all genera with multiple sampled species. The fragmentation of mecicobothriids into four separate lineages is strongly supported, with the genus Hexura nested within Antrodiaetidae. The three known species in the synonymized genus Atypoides form a clade sister to “traditional” Antrodiaetus species (Fig. 3), consistent with the well-supported 4-gene results of Hendrixson & Bond (2009), Figs. 1 and 2). Results of character evolution and divergence time analyses are presented and discussed below.

Discussion

Arachnid UCEs

We discovered that the arachnid bait set targets and recovers mostly exons, as suggested by Bossert & Danforth (2018) for arthropod UCE baits in general (see also Branstetter et al., 2017; Bossert et al., 2019 for hymenopterans). As such, arachnid UCE work is essentially exon capture, with flanking introns also captured for some loci. This of course has important implications for data analysis, because as we have shown here, this functional information can be used to refine analyses in various ways. Our finding also means that it might be possible to extract UCE loci from large spider/arachnid transcriptome datasets (e.g., Sharma et al., 2015; Garrison et al., 2016; Fernández et al., 2018), particularly at deeper phylogenetic levels where exon-only data would provide sufficient signal. Such a combined strategy was recently used in bee phylogenomics (Bossert et al., 2019).

Figure 3 Tree topology from RAxML concatenated analysis, based on filtered exon + intron matrix.

Support values for all nine analyses either indicated directly, or by circles at nodes (when exceeding 95 or 0.95 for all). Support values within (Namirea, Euagrus, Bymainiella, Calisoga) clade not shown, as these relationships vary across analyses (see .tre files in Data S1). Also, two low support nodes not shown for Phyluce unfiltered SVD results, as follows: Antrodiaetus apachecus + A. roretzi (67), Antrodiaetus hadros + A. gertschi (68).

Obviously, orthology is a fundamental premise in phylogenetic analyses. We found that the Phyluce unfiltered matrix included alignments with non-orthologs, confirmed via RAxML analysis. This “paralogy” persisted despite bioinformatic filters in place at both probe design (Faircloth, 2017) and Phyluce pipeline (Faircloth, 2016) stages. Our findings should not be taken as a criticism of these filters, because initial probe-design does not guarantee perfect orthology (Faircloth, 2017), and because we matched contigs to probes at liberal values (minimum coverage and minimum identity values of 65). Here we anticipated a tradeoff, as increasing this value would likely decrease non-orthology, but at the same time reduce the number of returned loci. Part of the issue is that the arachnid bait set was designed for sequence capture across all arachnids (Faircloth, 2017; Starrett et al., 2017), with a common ancestor that likely lived over 500 MYA (e.g., Rota-Stabelli, Daley & Pisani, 2013). Of all available UCE bait sets (e.g., amniotes, fish, various insects), this represents the greatest phylogenetic depth –the design of more taxon-specific bait sets within Arachnida, in combination with more stringent probe matching values is expected to largely (but probably not entirely) alleviate issues with non-orthology.

Empirical studies have shown that large phylogenomic datasets can be misled even when a minute fraction of loci include non-orthologs (e.g., Brown & Thomson, 2017; Gatesy et al., 2018). Here analysis of the Phyluce unfiltered matrix (with most characters but also non-orthologs) returned trees with the same branching topology within Atypoidea as for filtered matrices (Fig. 3). However, these trees vary somewhat in branch support (Fig. 3), but importantly produce maximum likelihood topologies that differ conspicuously in estimated branch lengths (measured in nucleotide substitutions per nucleotide site). For example, estimated IQ-TREE branch lengths derived from the Phyluce unfiltered matrix are 1.5-3X longer than those estimated from the 70% filtered exon + intron matrix (Fig. S1), with both matrices produced using the same GBLOCKS settings. Exon-only trees have even shorter branch lengths (.tre files in Data S1), but this comparison is confounded by removal of a different class of data (faster-evolving intron sites). To the extent that branch lengths influence downstream inferences (e.g., estimates of divergence times, lineage-through-time analyses, etc.), these differences in matrix filtering could have potential analytical impacts.

We discovered that some UCE loci treated as separate alignments actually represent exons of the same protein. Via annotation, we confirmed that the conserved regions of these separate alignments represented different exons of typically very large proteins. Although unknown for the taxa studied here, these exons are likely separated by very long introns (using the known short exon-long intron structure of spiders as models, Sanggaard et al., 2014). Inclusion of “duplicate” loci should not negatively impact concatenated phylogenomic analyses. But if the exons represent a single recombinational unit, then treating duplicate alignments as independent would violate analytical assumptions of coalescent-based analyses. Also, for population-level analyses relying upon SNPs from UCE loci (e.g., Derkarabetian et al., 2018b), many commonly-used downstream analyses assume no linkage and inclusion of duplicate loci would not be justified.

To summarize, we used custom annotation and manual checking of alignments to show that (1) core regions of arachnid UCEs represent exons, (2) non-orthology sometimes persists in UCE alignments, despite upstream bioinformatic filters, (3) some “separate” loci in the arachnid bait set represent different exons of the same protein (although separated by introns of unknown length). We argue that manual checking of alignments derived from an analytical pipeline remains important (see also Bossert et al., 2019 for another UCE example). Table S3 summarizes which UCE loci have been recovered in arachnid studies to date, and whether these loci are duplicates or potentially non-orthologous. This summary information could be used to further refine UCE analyses in arachnids, e.g., to manually adjust the published bait set to remove duplicates and paralogous loci, where non-orthology is unlikely to be rectified with more stringent probe match values. As has happened for almost all other UCE bait sets, the refinement of the arachnid set is an expected and natural outcome of knowledge gained through empirical study.

Atypoid phylogeny

We found strong support for the monophyly of Atypoidea (following Simon, 1892), based on a molecular phylogenetic sample with all described living genera. Our sample included the key genera Hexurella and Mecicobothrium, never previously sampled in a molecular phylogenetic analysis, and also included multiple early-diverging lineages from Avicularioidea (Hedin et al., 2018a; Opatova et al., 2019). The Atypoidea hypothesis was championed early (Chamberlin & Ivie, 1945; Coyle, 1971; Coyle, 1974) but ultimately fell out of favor as putative synapomorphies for the group were interpreted as plesiomorphies (Platnick, 1977; Gertsch & Platnick, 1979), and the original cladistic morphological analyses for mygalomorphs failed to recover this clade (Raven, 1985; Goloboff, 1993). However, at approximately the same time, Eskov & Zonstein (1990) argued for atypoid monophyly, and these ideas were later supported by early Sanger-based research (Hedin & Bond, 2006; Bond et al., 2012), although these molecular studies never included all described genera.

The presumed monophyly and placement of mecicobothriids is key in arguments regarding atypoid monophyly. Similar to early-diverging “diplurid” mygalomorphs, living mecicobothriid genera use elongate lateral spinnerets to build silken funnel-and-sheet webs. Platnick (1977) considered mecicobothriids to be more closely related to “diplurids” than to atypids or antrodiaetids, although he only examined Megahexura and Hexura. Similarly, Goloboff (1993) recovered mecicobothriids (scored as a single terminal) in an early-diverging grade with “diplurids”, but moving the root placement in his preferred phylogeny by one branch recovers atypoid monophyly. In this sense, both the exposed polyphyly of mecicobothriids (see below), and the phylogenomic placement of Hexurella and Mecicobothrium as ancient, early-diverging atypoids that closely straddle the primary division in mygalomorphs (Fig. 3), become centrally important in helping to understand past arguments over morphological homology and polarity. Proposed morphological polarities and diagnostic characters for all primary atypoid lineages are discussed below in the Taxonomy section.

Our phylogenomic results for all described meciobothriid genera convincingly confirm the non-monophyly of this family (Fig. 3). This result is consistent with prior molecular phylogenetic analyses that included Megahexura and Hexura, never recovered as sister taxa (Fig. 2). Mecicobothriid genera are actually morphologically heterogeneous, with each living genus displaying morphological apomorphies in somatic and genital morphology, particularly in female spermathecal morphology (see Gertsch & Platnick, 1979; Eskov & Zonstein, 1990, see below). Non-monophyly and ancient divergences also help to explain the vexing biogeographic disjunction (Hexurella, Hexura, Megahexura from the western US; Mecicobothrium from southern South America) observed for included genera. Both fossil-calibrated molecular clock estimates indicate that Hexurella and Mecicobothrium stem lineages were likely present during the Triassic, well before the fragmentation of Pangea (Fig. 4, Fig. S2).

Figure 4 Chronogram derived from Phylobayes analyses, estimated using calibration with minimum age for the root node of Atypoidea = 210 MYA.

HPD values in brackets. Results using the alternative calibration included as Fig. S2. Geological times from http://www.geosociety.org/documents/gsa/timescale/timescl.pdf.

Cryptic species, webs, parallel diversification

Many mygalomorph genera are relatively ancient, morphologically conserved, and dispersal-limited, traits which lead to cryptic speciation. Cryptic species are common in mygalomorphs (e.g., Bond et al., 2001; Castalanelli et al., 2014; Leavitt et al., 2015), and found in the atypoids that have been examined closely, antrodiaetids in particular (Hendrixson & Bond, 2007; Satler et al., 2011; Starrett et al., 2018). For example, the single described species Antrodiaetus riversi from central California is actually a complex of multiple cryptic species (Hedin, Starrett & Hayashi, 2013). Based on relative branch lengths recovered in phylogenomic analyses (Fig. 3), and estimated Cretaceous/early Tertiary ages for genera (Fig. 4, Fig. S2) we predict that cryptic species also occur in the Californian Megahexura, in Hexurella, and in Hexura from Oregon. Hexura is interesting in that the two described parapatric species are apparently ancient, perhaps similar to patterns seen in Ensatina oregonensis / picta salamanders from the rich mesic forests of Oregon (e.g., Kuchta et al., 2009).

Character reconstructions indicate rather unambiguously that the ancestral entrance web construct for Atypoidea is a funnel-and-sheet web (Fig. 5), with multiple entrance types derived from this state. Trapdoors in the antrodiaetid genus Aliatypus may have evolved directly from funnel-and-sheet webs, rather than from collapsible collars (contra Coyle, 1971). The well-supported placement of Hexura inside Antrodiaetidae (Fig. 3), as also found in the phylogenomic results of Opatova et al. (2019), is key in this character evolution inference.

Figure 5 Ancestral character reconstruction for entrance web constructs.

Proportional likelihood values for funnel-and-sheet web shown at internal nodes.

We also reconstructed a funnel-and-sheet web as the ancestral state for all mygalomorphs (Fig. 5). Our sample for avicularioids is small, but importantly, includes all key early-diverging lineages (Bond et al., 2012; Hedin et al., 2018a; Opatova et al., 2019). Using a much more comprehensive taxon sample, Opatova et al. (2019) also reconstruct the ancestral web for avicularioids as a funnel-and-sheet web. Many authors have discussed mecicobothriid and “diplurid” web similarities as an example of convergence, for example Gertsch (1949) stated that “the hind spinnerets of these spiders are greatly elongated …. probably an adaptation for spinning the sheet web, …illustrates how in widely unrelated creatures similar activities often lead to the production of similar morphological features”. Instead, our phylogenomic results indicate that the funnel-and-sheet, and elongate lateral spinnerets used to produce these webs, is likely the plesiomorphic condition in mygalomorphs. One caveat is that our funnel-and-sheet scoring may be an over-simplification of homology for these taxa. For example, many early-diverging “diplurids” build massive sheet-like space webs that serve to capture prey (Coyle, 1986), features not obviously present in early-diverging atypoid webs.

Atypoid taxonomy

Here we summarize the revised taxonomy of Atypoidea and all included families, focusing on extant taxa (summarized in Fig. 6). The composition of the family Mecicobothriidae is revised. Megahexura and Hexurella are removed from Mecicobothriidae and each included in new families, while Hexura is transferred to the family Antrodiaetidae. Also within Antrodiaetidae, the genus Atypoides is formally removed from synonymy with Antrodiaetus. All nomenclatural changes proposed are to be attributed to Hedin and Bond.

Figure 6 Summary of new taxonomy and diagnostic morphological characters.

See text for references and explanation of terms.

The electronic version of this article in Portable Document Format (PDF) will represent a published work according to the International Commission on Zoological Nomenclature (ICZN), and hence the new names contained in the electronic version are effectively published under that Code from the electronic edition alone. This published work and the nomenclatural acts it contains have been registered in ZooBank, the online registration system for the ICZN. The ZooBank LSIDs (Life Science Identifiers) can be resolved and the associated information viewed through any standard web browser by appending the LSID to the prefix http://zoobank.org/. The LSID for this publication is: urn:lsid:zoobank.org:pub:A7E6FD73-9D49-4B55-911F-5D105B09A52C. The online version of this work is archived and available from the following digital repositories: PeerJ, PubMed Central and CLOCKSS.

Family Hexurellidae (NEW FAMILY) (urn:lsid:zoobank.org:act:504C322E-8FAC-4E25-806C-DCE37372112E)

Type genus. Hexurella (Gertsch & Platnick, 1979 urn:lsid:nmbe.ch:spidergen:00010) (type species H. pinea Gertsch & Platnick, 1979)

Diagnosis. As a consequence of its monogeneric status, characters used to diagnose Hexurellidae are those characters also attributed to the type genus Hexurella, as follows: (1) males having a gently coiled embolus, not corkscrew shaped (illustrated by Gertsch & Platnick, 1979, figures 77, 84, 87, 90); (2) posterior lateral spinnerets with four segments; and (3) spermathecae composed of a single bursal opening branching into four short, and relatively thicker bulbs (Gertsch & Platnick, 1979, figure 79). Conversely, megahexurid taxa appear to have much thinner spermathecal bulbs in which pairs share a bursal opening. As is the case for other new taxa and ranks proposed below, a more thorough examination of this new family’s morphology will be an important next step in diagnosing these groups.

Distribution. Distributed in upland habitats of southern California, northern Baja California, and central/southern Arizona (Gertsch & Platnick, 1979). Undescribed species likely occur in the mountains of northern Sonora, Mexico.

Family Mecicobothriidae Holmberg, 1882 (urn:lsid:nmbe.ch:spiderfam:0003) (new circumscription)

Type genus. Mecicobothrium Holmberg , 1882 (urn:lsid:nmbe.ch:spidergen:00011) (type species Mecicobothrium thorelli Holmberg, 1882)

Diagnosis. Characters used to diagnose the family are those characters attributed to the type genus. Adult males of described species have a long and distinctly coiled corkscrew-shaped palpal embolus (e.g., Gertsch & Platnick, 1979, figures 45, 48, 49; Lucas et al., 2006, figures 1–3) that distinguishes members of this family from all other atypoid taxa. Males also have a unique anterior cheliceral apophysis (Gertsch & Platnick, 1979, figures 40–42; Lucas et al., 2006, figures 20–21). Females have distinct spermathecal bulbs comprising four receptacles with the outer pair much shorter and rounder than the inner two (Gertsch & Platnick, 1979, figure 38); we note that females of M. baccai are unknown.

Distribution. The two described species are known from Argentina, Uruguay, and Brazil.

Family Megahexuridae (NEW FAMILY) (urn:lsid:zoobank.org:act:0D009AAF-B71C-4FFA-A580-DCD67BAA48AB)

Type genus. Megahexura Kaston, 1972 (urn:lsid:nmbe.ch:spidergen:00012)

(type species Hexura fulva Chamberlin, 1919)

Diagnosis. Characters used to diagnose the family Megahexuridae are those attributed to the type genus. Members of this family can be diagnosed from other atypoid taxa by having a carapace with expanded pleurites at the posterior lateral corners (Gertsch & Platnick, 1979 figures 51, 53). Megahexurid females have spermathecae with four thin elongate bulbs, with a single receptacle opening for each pair (Gertsch & Platnick, 1979, figure 57).

Distribution. The single described species (M. fulva) is known from upland habitats of southern and central California (Gertsch & Platnick, 1979), although populations likely occur in northern Baja California. Megahexura fulva likely includes cryptic species (Fig. 4, Fig. S2).

Family Antrodiaetidae Gertsch,1940 (urn:lsid:nmbe.ch:spiderfam:0002)(new circumscription)

Type genus. Antrodiaetus Ausserer, 1871 (urn:lsid:nmbe.ch:spidergen:00007)

(type species Antrodiaetus unicolor Hentz, 1842)

List of included genera AliatypusSmith, 1908 (urn:lsid:nmbe.ch:spidergen:00006)	
AntrodiaetusAusserer, 1871 (urn:lsid:nmbe.ch:spidergen:00007)	
HexuraSimon, 1884 (urn:lsid:nmbe.ch:spidergen:00009)	

Atypoides O. Pickard-Cambridge, 1883. (type species Atypoides riversi O. Pickard-Cambridge, 1883 by monotypy). Here formally removed from synonymy of Antrodiaetus Ausserer, 1871 contra Hendrixson & Bond (2007: 752).

List of included species

Atypoides riversi O. Pickard-Cambridge, 1883	
Atypoides hadrosCoyle (1968)	
Atypoides gertschiCoyle (1968)	

Diagnosis. Adult male antrodiaetids possess a palpal bulb with a branched conductor, with inner and outer conductor sclerites (following Coyle, 1971, figure 325). The possession of this character state in Hexura was noted in the addendum of (Eskov & Zonstein, 1990), based on observations of Dr. F. Coyle, and confirmed by our study of male Hexura specimens.

Following Coyle (1968), the genus Atypoides can be distinguished from Antrodiaetus in having three pairs of spinnerets (Coyle, 1968, figure 30–32), with adult males possessing cheliceral apophyses (Coyle, 1968, figure 46–52). Many features separate Atypoides and Antrodiaetus from Hexura and Aliatypus.

Distribution. Aliatypus and Hexura are known from the western United States (Coyle, 1974; Gertsch & Platnick, 1979), Atypoides is from the western US and the southern Ozarks (Coyle, 1968; Hedin, Starrett & Hayashi, 2013), while Antrodiaetus includes species in Japan and more broadly in North America (Coyle, 1971; Hendrixson & Bond, 2007). Cryptic species are likely in all four genera.

Comments. Although megahexurids are sisters to antrodiaetids, we do not place them in the same family for three primary reasons. First, these families share a common ancestor that likely existed over 200 million years ago (Fig. 4). This level of divergence would exceed any intra-familial divergence in described mygalomorph families (see Opatova et al., 2019). Second, these families differ in important diagnostic characters, including female spermathecal morphology, but importantly megahexurid males lack the key antrodiaetid palpal bulb with diagnostic inner and outer conductor sclerites (Fig. 6).

Conversely, one could argue that Aliatypus and Hexura each deserve family-level status (the latter an available family rank name, Hexurinae Simon 1889), sister to other antrodiaetids. Again, although heterogenous from a web construct perspective (Fig. 5), antrodiaetids share morphological synapomorphies, with a level of inter-generic temporal divergence comparable to other described mygalomorph families (Fig. 4, Opatova et al., 2019).

Conclusions

Early-diverging atypoid lineages are ancient, often species-poor (approximating monotypic), and use silk to build funnel-and-sheet webs. The evolution of more diverse silken entrance constructs is found in more derived atypoid lineages. Similar patterns of species-poor early-diverging lineages, and diverse entrance constructs evolving in more derived lineages occurs in parallel in the avicularioid mygalomorphs (Opatova et al., 2019). In this sense, atypoids and avicularioids represent comparable evolutionary experiments, although the latter clade has clearly evolved a greater diversity of taxa, morphologies, and web constructs. How the competitive interplay of these parallel lineages has impacted diversification dynamics in deep time would be an interesting topic for further study.

Supplemental Information

Table S1 Voucher specimen information, input DNA values, number of assembled contigs, and number of UCE loci per taxon for each of three matrices

Click here for additional data file.

Table S2 UCE annotations based on custom BLAST to Parasteotoda, Stegodyphus, Ixodes and Limulus (as described in text)

Categories for each locus (e.g., duplicate protein, paralogy evident, etc.) grouped by cell color as explained in file.

Click here for additional data file.

Table S3 Summary of UCE loci for published arachnid datasets and information for custom BLAST to Parasteotoda.

Sheet #1 includes summary of UCE loci for published arachnid datasets. Sheet #2 includes custom BLAST to Parasteotoda only.

Click here for additional data file.

Figure S1 IQ-TREE maximum likelihood results

(A) Phyluce unfiltered and (B) “Filtered 70% Exon + Intron” matrices.

Click here for additional data file.

Figure S2 Chronogram derived from alternative Phylobayes analysis

Estimated using calibration with minimum age for the root node of common ancestor of Atypidae < > Antrodiaetidae at 100 MYA, as described in text.

Click here for additional data file.

Data S1 Input .phy files, and resulting .tre files, as described in text

Click here for additional data file.

We thank Nobuo Tsurusaki, Dean Leavitt, Jordan Satler and in particular Jim Starrett for help with specimen collection. Gabriel Pompozzi provided images of Mecicobothrium. Comments of Brent Hendrixson, D. Leavitt, J. Starrett and an anonymous reviewer helped to improve the manuscript.

Additional Information and Declarations

Competing Interests

Author Contributions

Field Study Permissions

DNA Deposition

Data Availability

New Species Registration

The authors declare there are no competing interests.

Marshal Hedin and Shahan Derkarabetian conceived and designed the experiments, performed the experiments, analyzed the data, contributed reagents/materials/analysis tools, prepared figures and/or tables, authored or reviewed drafts of the paper, approved the final draft.

Adan Alfaro conceived and designed the experiments, performed the experiments, analyzed the data, approved the final draft.

Martín J. Ramírez and Jason E. Bond contributed reagents/materials/analysis tools, authored or reviewed drafts of the paper, approved the final draft.

The following information was supplied relating to field study approvals (i.e., approving body and any reference numbers):

Australian collections were made under permits from Queensland (Permit Number WISP01242003). Specimens were exported under permit from Environment Australia.

The following information was supplied regarding the deposition of DNA sequences:

SRA, BioProject ID:PRJNA517633: http://www.ncbi.nlm.nih.gov/bioproject/517633.

The following information was supplied regarding data availability:

SRA (SAMN10839235 - 10839249).

The following information was supplied regarding the registration of a newly described species:

Publication LSID urn:lsid:zoobank.org:pub:A7E6FD73-9D49-4B55-911F- 5D105B09A52C

Family name: Hexurellidae LSID urn:lsid:zoobank.org:act:504C322E-8FAC-4E25-806C-DCE37372112E

Family name: Megahexuridae LSID urn:lsid:zoobank.org:act:0D009AAF-B71C-4FFA-A580-DCD67BAA48AB.

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
