# Peer review of "Phylogenomic analysis and revised classification of atypoid mygalomorph spiders (Araneae, Mygalomorphae), with notes on arachnid ultraconserved element loci"

_PeerJ, doi:10.7717/peerj.6864_

## Round 0.1 · original submission · Minor Revisions

Dear Dr. Hedin and colleagues:

Thanks for submitting your manuscript to PeerJ. I have now received two independent reviews of your work, and as you will see, both are very favorable. Well done! Nonetheless, the reviewers raised some relatively minor concerns about the research, and areas where the manuscript can be improved. I agree with the reviewers, and thus feel that their concerns should be adequately addressed before moving forward.

Aside from the criticisms raised by the reviewers in their reports, be sure to thoroughly evaluate the marked-up manuscripts kindly provided by both reviewers.

Therefore, I am recommending that you revise your manuscript accordingly, taking into account all of the issues raised by the reviewers. I do believe that your manuscript will be ready for publication once these issues are addressed.

Good luck with your revision,

-joe

Reviewer 1 ·

Basic reporting

Since I’m not an English native speaker, I did not make any changes regarding the grammar. The manuscript is well written, clear and technically correct.

The introduction presented a background of previous phylogenetic evidences on the study clade of Mygalomorphae and also discussed about the use of ultraconserved elements in phylogenetic analyses. Although the background is complete, the introduction could include some information about biogeographic history of the clade, or comments about the genera distributions.

Figures are appropriate and the supplementary material provided is very complete and informative.

Experimental design

The work is well-designed and the chance to analyze specimens of particular genera that were not previously studied adds a relevant feature. Methods are described with a good level of detail, just some details on general comments about this section are presented below.

Validity of the findings

The data of the manuscript which the conclusions are based are available through various repositories. The conclusions are related with the main questions of the work, and are discussed including updated references. The findings are of great interest, since some taxonomic changes at family levels are proposed in a relevant clade of spiders. This work will promote new studies to develop that could understand the evolution of some clades.

Additional comments

Introduction

Lines 65–66. Maybe you can add some references to this statement, by adding some of the works related with theraposid phylogeny (e.g. Hamilton et al. 2011, Guadanucci 2014, Ortiz et al. 2018).
Lines 73–74. Explain with more detail this statement.
Lines 80 – 82. Add some references about the morphological characterizations of Atypoids, for example, Gertsch & Platnick (1979).
Line 100. Please update the reference to “World Spider Catalog (2019)”.
Lines 130–132. This sentence is long and little confusing, try to rewrite it.

Methods

Line 137. Update the reference to 2019.
Line 139. You stated “…(species now included in Antrodiaetus)…”. I assume that this is not a result from the present work, thus include a reference.
Line 140. Which means by “traditional”? please explain better.
Line 145. Samples of Mecicobothrium, Calommata and Atypus were represented by single specimens. Could you state to which species corresponded the materials?
Line 213. The unpartitioned analysis were conducted with the GTR + gamma model. This was achieve by using the ModelFinder, as stated below?

Discussion

Lines 387–388. Add the reference of Goloboff (1993).

Figures

It would be great if authors could include a figure of a member of Mecicobothrium in Figure 1.

Figure 2. I suggest to present the different phylogenetic hypotheses from previous works in a chronological order, thus it is easy to follows.

·

Basic reporting

no comment

Experimental design

no comment

Validity of the findings

MS# 34648 Review of Phylogenetic analysis and revised classification of atypoid mygalomorph spiders (Araneae, Mygalomorphae), with notes on arachnid ultraconserved element loci

In this manuscript, Marshal Hedin and colleagues employ UCE loci for inferring the phylogeny of atypoid mygalomorph spiders, revise the group’s taxonomy, comment on the evolution of these spiders (including divergence dating and silk use), and discuss the utility of UCE loci in arachnid phylogenomics. The paper is clear and well researched and is an important contribution to our understanding of the systematics and evolution of the Atypoidea, a group with a contentious phylogenetic and taxonomic history.

The discussion of character evolution and divergence times are supported by rigorous analyses and do not require additional commentary. My remarks primarily deal with the taxonomic section:

• Mecicobothriidae is polyphyletic and requires taxonomic revision. The authors propose to solve this by making Mecicobothriidae a monotypic family; creating two new families to accommodate Hexurella and Megahexura; and transferring Hexura to Antrodiaetidae. While I do not disagree with any of the proposed taxonomic changes, I do wonder how the authors would have treated Hexura had it been placed sister to the other antrodiaetids rather than nested within them. Would the authors have named a new family (“Hexuridae”) or would they have still considered it an antrodiaetid based on shared palpal bulb morphology? I bring this point up because the classic antrodiaetids (Aliatypus + Antrodiaetus/Atypoides) can be considered adaptively/ecologically divergent (trapdoors versus collars/turrets) and I suppose it can be argued that the two lineages perhaps warrant family status on their own accord (e.g., “Aliatypidae”). Similarly, could Megahexura – although ancient and quite distinct – not now be considered an antrodiaetid given its phylogenetic placement? The authors should be more explicit on what they consider an atypoid spider family and discuss the factors they consider most important for family-level recognition (e.g., morphological apomorphies, adaptive divergence, age, nomenclatural stability, etc.) so that the taxonomic conclusions are repeatable.

• The three antrodiaetid species formerly placed in Atypoides (= Antrodiaetus) form a monophyletic group sister to the “original” Antrodiaetus – and as a result, the authors resurrect Atypoides (contra Hendrixson & Bond, 2007). This change is justified based on the inclusion of Antrodiaetus roretzi (a species spanning the root node of the genus and who shares a plesiomorphic palpal bulb morphology with Atypoides). Given this change, the authors should also provide a revised diagnosis for Antrodiaetus and summarize the diagnostic morphological characters for the genus in Figure 6. The authors may consider including a reference to Hendrixson & Bond (2009) who also found strong support for Atypoides (in at least some analyses) but did not call for any taxonomic changes.

• For the sake of consistency and efficacy, the authors should consider making reference to specific figures of diagnostic features in the diagnoses for each revised taxon (as is done for Megahexuridae, e.g., Line 508).

---

## Round 0.2 · accepted · Accept

Dear Dr. Hedin and colleagues:

Thanks for revising your manuscript based on the minor concerns raised by the reviewers. I now believe that your manuscript is suitable for publication. Congratulations! I look forward to seeing this work in print, and I anticipate it being an important resource for the groups studying atypoid mygalomorph systematics. Thanks again for choosing PeerJ to publish such important work.

Best,

-joe

#